# PLANNING WITH AN ENSEMBLE OF WORLD MODELS

## ABSTRACT

Motion planning is of critical importance for safe navigation in complex urban environments. Historically, motion planners (MPs) have been evaluated using procedurally-generated simulators like CARLA. However, such synthetic benchmarks are not reflective of real-world multi-agent interactions. nuPlan, a recently released MP benchmark, addresses this limitation by augmenting real-world driving logs with closed-loop simulation logic, effectively turning the fixed dataset into a reactive "gym" simulator. We evaluate the quality of nuPlan's Default-Gym and find that it does not accurately reflect real-world human behavior, particularly for cities with unique driving behaviors (e.g., Boston drivers tend to be more aggressive than Pittsburgh drivers). Therefore, we propose city-specific gyms (e.g., a Boston-Gym and Pittsburgh-Gym) to evaluate planning performance. Evaluating a state-of-the-art planner with our proposed ensemble of gyms yields a drop in performance, suggesting that a good planner must adapt to different environments. Leveraging this insight, we present City-Driver, a model-predictive control (MPC) based planner that unrolls a city-specific world model that adapts to different driving conditions. Our extensive experiments demonstrate that City-Driver achieves state-of-the-art results on the nuPlan benchmark, reducing test error from 6.4% to 4.8%.

## 1 INTRODUCTION

Motion planning (MP) is a critical component of the autonomy stack. Autonomous Vehicles (AVs) must carefully plan their motion to navigate in complex urban environments to safely reach their goal destination, avoid collisions, and abide by the rules of the road. Motion planners are typically trained and evaluated in synthetic environments like CARLA (Dosovitskiy et al., 2017) and AirSim (Shah et al., 2017). However, such simulated environments notoriously suffer from a sim-to-real gap due to systematic biases and a lack of real-world diversity. Instead, we focus on the recent nuPlan planning benchmark (Caesar et al., 2021), which makes use of *real* driving logs to create environments for training and evaluation. Such data-driven simulation is a core enabling technology in current autonomy stacks through the use of *re*simulation and log playback (Caesar et al., 2021). We believe that such data-driven reactive benchmarks may represent a watershed moment in the development of motion planners, *provided* that one can trust the world simulator.

Although nuPlan evaluates algorithms on ego-centric forecasting accuracy (defined as the C1 metric (Caesar et al., 2021)) and motion planning in a non-reactive world (C2), our work focuses on motion planning in a reactive world (C3) as this most closely resembles real-world deployment. Recent methods like PDM-C (Dauner et al., 2023) have dramatically improved planning performance in reactive environments by combining model-predictive control (MPC) with classical planning algorithms like the Intelligent Driver Model (IDM) (Treiber et al., 2000).

To better understand PDM-C, we visualize its performance in nuPlan's reactive simulator. Surprisingly, we find that nuPlan's default evaluation environment (which we refer to as the Default-Gym, akin to simulation environments used in reinforcement learning) dramatically differs from the recorded behavior of human drivers. Importantly, nuPlan's Default-Gym endows agents with unrealistic behaviors and unrealistically over-populates the drivable region (cf. Fig 2). As shown in Figure 1, simulated agents have smaller distance to the ego-vehicle in the Default-Gym than in real-world driving logs. This is the primary reason for the surprising fact that the ground-truth human motion plan achieves only an accuracy of 93.12% under the Default-Gym, as reported in Dauner et al. (2023).

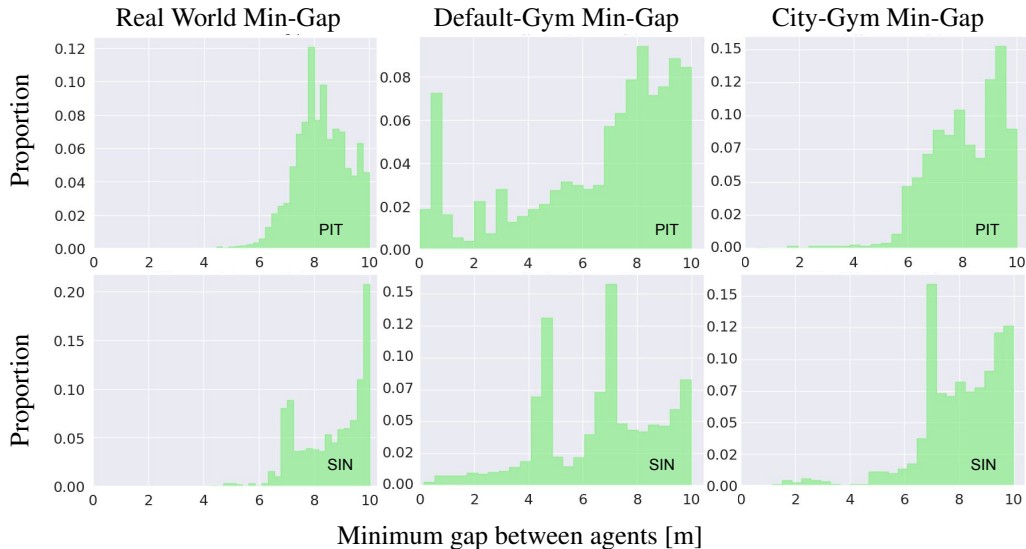

Figure 1: **Per-City Behavior Statistics.** We compare the distance (min gap) between the ego-vehicle and lead agent in PIT (Pittsburgh) and SIN (Singapore) using three different simulators. We plot the real-world distribution on the left, Default-Gym in the center and city-specific gym on the right. We find that the min-gap distribution in our proposed city-specific gyms closely model real-world behavior statistics. In contrast, the Default-Gym has significantly different distribution.

This raises the question; how do we find the "right" simulator? We posit that a realistic simulator should closely mimic recorded agent behavior when the ego-vehicle acts exactly as in the original driving log. This suggests that human performance can be used as a *metric* for evaluating the quality of a simulator. By optimizing agent behaviors for this metric, we can build a more realistic gym that improves realism (as measured by human performance across the entire dataset). Moreover, we find that optimizing *within* city-specific datasets further improves realism, allowing us to effectively build city-specific gyms. For example, agents in our Boston-Gym drive more aggressively (compared to Pittsburgh) by accelerating faster and maintaining a smaller gap to the lead vehicle.

Armed with our city-specific gyms, we can now evaluate the SOTA planners (such as PDM-C) across an ensemble of gym environments. We notice a considerable drop in performance. Unsurprisingly, PDM-C's model of the world was tuned for the Default-Gym. Recall that MPC produces a motion plan by internally unrolling a world model and optimizing for a plan that minimizes a cost function in that world. One can improve accuracy by simply using the appropriate City-Gym when unrolling. We refer to MPC-based planners that use City-Specific world models as City-Drivers. Intuitively, a Boston-Driver should outperform the default PDM-C in the Boston-Gym. However, since the nuPlan benchmark does not provide access to such city information, we learn a city classifier to adaptively select PDM-C's driver model based on city-specific driving characteristics. City-Driver, our simple model-predictive control (MPC) based planner, achieves state-of-the-art results on the nuPlan benchmark.

**Contributions.** We present three major contributions

- We demonstrate that nuPlan's Default-Gym does not accurately reflect real-world human behavior. We quantify the realism of C3 and propose city-specific alternatives.

- We present City-Driver, an adaptive planner based on model-predictive control (MPC) that unrolls and executes city-specific driver models to safely navigate in diverse scenarios.

- We conduct extensive experiments to ablate our design choices and demonstrate that our simple method achieves state-of-the-art results on the nuPlan benchmark.

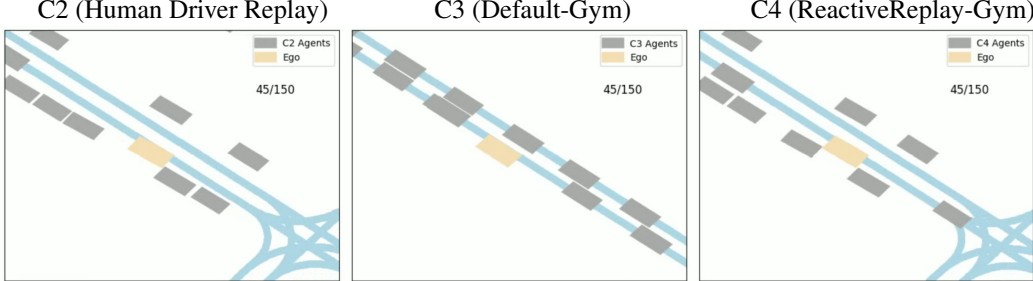

Figure 2: We visualize three simulation settings (Human Driver Replay, Default-Gym, and ReactiveReplay-Gym). We visualize a driving log from PIT at timestep=45 (total=150 timesteps). The ego vehicle is shown in Yellow, other agents are shown in Gray, and the road graph is shown in Blue lines. We notice that the Default-Gym over-populates the scenario by projecting parked cars onto the road graph. This significantly differs from the recorded agent behaviour (as seen in the human driver replay). As a result, human planning performance drops in the Default-Gym (cf. Table 3). We address this issue with our proposed ReactiveReplay-Gym, which allows agents to follow their ground-truth trajectory while remaining reactive to other agents.

## 2 RELATED WORKS

**Rule-Based Planning.** Although recent work focuses on learning robust policies by predicting goal-conditioned way-points, cost-volumes, and reward functions, rule based planners still outperform learning-based approaches on real data (Dauner et al., 2023). Rule-based planners are well studied (Stentz, 1994; LaValle & Kuffner Jr, 2001; Reeds & Shepp, 1990; González et al., 2015; Zhou et al., 2022), and have been widely adopted due to their safety guarantees and interpretability (Thrun et al., 2006; Bacha et al., 2008; Leonard et al., 2008; Urmson et al., 2008; Chen et al., 2015). Given the current longitudinal position, velocity, and distance to the leading vehicle, rule-based planners estimate longitudinal acceleration to safely progress towards the goal state. The Intelligent Driver Model (IDM) (Treiber et al., 2000) is a classic non-learned algorithm for vehicle motion planning that relies on graph-based search to reach a goal while employing a PID velocity controller to avoid collisions with other vehicles. Dauner et al. (2023) extends the IDM by sampling multiple trajectories and unrolling a world model to select an optimal trajectory with minimum cost (*e.g.* time to collision). This improves collision avoidance without resulting in overly conservative motion plans.

**Trajectory Optimization**. Motion planning is often framed as an optimization problem of a hand-designed cost function, which is then minimized to generate the desired trajectory Buehler et al. (2009); Montemerlo et al. (2008); Fan et al. (2018); Ziegler et al. (2014). To simplify this process, cost functions assume a quadratic objective function or divide the planning task into its lateral and longitudinal components. Approaches such as A* Ajanovic et al. (2018), RRT Karaman & Frazzoli (2011), and dynamic programming Fan et al. (2018) are commonly used to search for optimal solutions. Phan-Minh et al. (2020) generates a set of trajectories and evaluates them based on a predefined cost, selecting the trajectory with the lowest cost. While these methods are attractive due to their parallelizability, interpretability and functional guarantees, they are not robust when applied to real-world driving scenarios and require significant effort to fine-tune.

Conventional trajectory optimization approaches typically aim to compute a complete trajectory that spans from the initial configuration to the desired goal configuration. However, given the inherently dynamic and uncertain nature of the driving environment, precise long-horizon motion plans cannot be predicted in advance. Model-predictive control (MPC) has gained prominence in recent years for real-time path planning (Rastelli et al., 2014; LaValle, 2006; Karaman & Frazzoli, 2010; Pongpunwattana & Rysdyk, 2004) because it adopts an iterative cost minimization strategy to select a locally optimal trajectory for each timestep. This allows MPC-based algorithms to adapt to changes in the environment.

**Data-Driven Simulation**. In recent years, many machine learning-based motion planners have emerged, leveraging the availability of simulator environments like CARLA (Dosovitskiy et al., 2017), AirSim (Shah et al., 2017), and others. However, current simulators are limited because they

| Evaluation Type | Ego Simulation | Agent Simulation | Agent Path | Gym |
|---|---|---|---|---|
| Log Replay (C1) | Open loop | Open loop | GT | None |
| Closed Loop (non-reactive) (C2) | Closed loop | Open loop | GT | None |
| Closed Loop (reactive, C3) | Closed loop | Closed loop | Road graph | Default |
| Closed Loop (reactive, C4) | Closed loop | Closed loop | GT | ReactiveReplay |

Table 1: **nuPlan Evaluation Setup**. The nuPlan benchmark introduces three evaluation environments (C1, C2 and C3). Open loop evaluation implies the all agents (including the ego-vehicle) follow their ground truth trajectory (*e.g.* world on rails). In contrast, closed loop evaluation means the agents must plan and act in the simulated environment. Visualizing the Default-Gym reveals that all agents (including parked cars) are projected onto the lane-graph leading to an overpoluated drivable region. This dramatically differs from the human driver replay as shown in Figure 1 and 2. Therefore, we introduce the ReactiveReplay-Gym which allows agents to traverse along their GT trajectory while also reacting to surrounding vehicles.

rely on synthetic data generated from video game engines and have insufficient visual fidelity. Importantly, they lack the necessary diversity of driving scenarios required for comprehensive training and evaluation. More recently, Montali et al. (2023) introduce the Waymo sim agents challenge which evaluates simulators by comparing the accuracy of all agents against their ground-truth trajectories. Furthermore, CommonRoad (Althoff et al., 2017) offers a driving dataset and a planning benchmark with partially collected real-world data and partly hand-crafted.

In contrast, nuPlan (Caesar et al., 2021) augments real-world driving logs with closed-loop simulation logic, effectively turning the fixed dataset into a reactive simulator. nuPlan has released 1300 hours of real world driving logs from various cities including Las Vegas, Boston, Pittsburgh, and Singapore. Driving in each city presents its unique set of driving challenges. For example, Las Vegas has many high density pick-up and drop-off locations, and intersections with 8 parallel driving lanes per direction. In Boston, drivers tend to double park, creating distinct route challenges.

## 3 EVALUATION WITH A CITY-SPECIFIC GYM

nuPlan (Caesar et al., 2021) evaluates motion planners using a data-driven reactive world model (which we refer to as a Gym). Simulated agents are endowed with rule-based planning policies that react to the ego-vehicle's plan. In this section, we discuss the limitations of nuPlan's Default-Gym and describe our proposed modifications to improve realism of nuPlan's reactive world model.

**nuPlan Evaluates Motion Planners with Reactive Simulators.** The nuPlan dataset releases 1300 hours of real-world driving logs. From these driving logs, nuPlan carefully curates thousands of 15-seconds clips to capture interesting scenarios (e.g. unprotected left turns). These driving logs are used to evaluate both open-loop and closed-loop planning performance. nuPlan evaluates open-loop performance with ego-centric forecasting accuracy (C1). nuPlan offers closed-loop simulation with planning evaluation in a non-reactive world where all agents except the ego-vehicle replay their ground-truth trajectory (C2), and planning evaluation in a reactive world (C3) where all agents react to the ego-vehicle using rule-based planners (e.g. IDM (Treiber et al., 2000)). Our work focuses on C3 since this most closely resembles real world deployment. The IDM planner used for all other agents is initialized using hyperparameters ($\theta_i, \forall i \in \{0, 1, 2, 3, 4\}$) for the target velocity, minimum gap, headway time, maximum acceleration, and maximum deceleration. These parameters allow the IDM to simulate realistic driving behavior.

**Reactive Gyms are Realistic Simulators**. Ideally, motion planners should be evaluated on real vehicles in-the-wild. However, testing planning algorithms in densely populated areas (where most challenging planning scenarios occur) is prohibitively dangerous. Therefore, we opt to use a data-driven reactive simulator. We posit that reactive simulators are a good proxy for real-world driving performance, *provided* that we can trust the simulated environment.

**How Can We Quantify the Realism of a Gym?** A realistic simulator should closely mimic recorded agent behavior when the ego-vehicle acts exactly as in the original driving log. To quantify the *realism* of a gym, we evaluate the performance of human drivers (using recorded logs) in the given gym. To our surprise, the ground-truth human motion plan only achieves an accuracy of

93.12% in nuPlan's Default-Gym (C3) as shown in Table 3. In Figure 2, we note that simulated agent behaviours (subfigure b) are dramatically different from ground-truth agent behavior (subfigure a). This is because all agents in nuPlan's Default-Gym are snapped onto the road graph. Consequently, this unrealistically overpopulates the road, leading to lower human driver performance.

**Limitations of the Default-Gym**. To address the above limitation of the Default-Gym (C3), we allow simulated agents to traverse on the ground-truth trajectory without snapping to the road graph (C4) as shown in Figure 2(c). Unsurprisingly, this improves the human performance to 93.52% (cf. Table 3). However, there is still room for improvement.

**Each City Needs Its Own Gym**. We note that nuPlan's Default-Gym uses a single world model (defined by a single set of IDM hyperparameters $\theta^{DEF}$) that control agent behaviors across all cities in the dataset. As a result, the evaluation protocol assumes that all agents behave similarly across cities. However, we find that this assumption does not hold. As shown in Figure 3, Boston drivers maintain a smaller gap to the leading vehicle when compared to Pittsburgh drivers, motivating the need for city-specific gyms. We optimize agent behaviours ($\theta^{CITY}$) for each city using human accuracy as our objective function,

$$\theta^{CITY} = \max_{j \in S} M(H, W(\theta_0^j, \theta_1^j, \theta_2^j, \theta_3^j, \theta_4^j)) \tag{1}$$

where $H$ is the human planner, $W$ is the IDM-based world model and $S$ is the set of all possible parameters for $\theta$. We optimize $\theta^{CITY}$ using grid-search over $S$. We tabulate the hyperparameters for each city-specific gym in Table 4.

Using this ensemble of city-specific gyms, we can evaluate the performance of planners in a variety of environments. In addition to simulating per-city driving characteristics, we can simulate out-of-distribution behaviors. For example, we can simulate Boston-like driving behavior (using the Boston-Gym) for driving logs collected in Pittsburgh.

## 4  PLANNING WITH A CITY-SPECIFIC DRIVER

In this section, we provide relevant background on PDM-C (Dauner et al., 2023). Further, we discuss how to leverage insights about city-specific gyms to improve MPC-based planners.

**PDM-C is Actually An MPC-based Planner**. PDM-C is a state-of-the-art rule-based planner that improves upon the Intelligent Driver Model (IDM). Recall that the IDM is a car following model that employs a simple longitudinal PID velocity controller along a given reference path. PDM-C converts the IDM to an MPC-based planner by internally unrolling a world model and returning a plan that minimizes a cost function over that world model. PDM-C generates a set of IDM path proposals by modulating longitudinal velocities and lateral offsets to the reference path and scores each proposals based on its internal world model. The path proposal with the highest score is executed. The scoring function uses metrics similar to the Default-Gym (e.g. time to collision, ego progress to goal) while a constant velocity forecast is used as the internal world model. We refer readers to Dauner et al. (2023) for further details.

**Improving PDM-C's World Model**. We evaluate PDM-C's performance on the Default-Gym and our proposed ensemble of city-specific gyms in Table 3 and Table 6. Unsurprisingly, we notice a drop in performance from 93.61 to 92.90 in C3 and 93.02 to 91.83 in C4. We posit that because PDM-C's internal world model (which we refer to as a driver model) is a non-reactive, city-agnostic constant velocity forecaster that is well-tuned for the Default-Gym. Instead of relying on a non-reactive constant velocity world model, we can use a city-specific reactive world model (similar to that used for evaluation in C3 or C4). Specifically, when evaluating PDM-C in the Boston-Gym, we expect that using a Boston-Driver will improve planning performance. We find the optimal values of the IDM hyperparameters ($\theta^{CITY}$) using Equation 1.

**City Identification for Adaptive Planning**. To accurately identify the right driver model to use with PDM-C, we need city-level information. However, since the nuPlan benchmark does not provide this information, we learn a classifier to identify the city from local maps. We use LaneNet (Caesar et al., 2021) which is inspired by LaneGCN (Liang et al., 2020). LaneNet takes a vectorized road graph of radius $R$ around the ego-vehicle as input and predicts the city. The network consists of

| Models (Dauner et al., 2023) | City | C1 (Open Loop) | C2 (Non-Reactive) | C3 (Reactive) |
|---|---|---|---|---|
| PDM-O | All | **80.01** | 77.70 | 79.36 |
| PDM-C | All | 67.23 | 94.49 | 93.61 |
| PDM-H | All | 79.50 | **94.49** | **93.61** |

Table 2: **Baseline Performance.** We show SoTA methods (PDM-O, PDM-C, PDM-H) achieve high performance across C1, C2, C3 on the nuPlan test set. Note that PDM-C and PDM-H produces the same accuracy for C3, since they both execute the same short-horizon plan. As such, we focus on C3 accuracy since that evaluates planners in a true reactive world, provided that one can trust the world simulator.

| Models | City | C3 (Default-Gym) | C4 (ReactiveReplay-Gym) |
|---|---|---|---|
| PDM-C (Dauner et al., 2023) | All | 93.61 | 93.02 |
| Human Replay Planner | All | 93.12 | 93.52 |

Table 3: **Default-Gym vs ReactiveReplay-Gym**. Recent SoTA methods (Dauner et al., 2023) have dramatically improved performance by leveraging model predictive control (MPC) methods that evaluate the goodness of potential actions by unrolling a world model. We also evaluate human performance, but curiously find it underperforms SoTA for a reactive world (C3). We find the C3 simulator is unrealistic because it over-populates the drivable region (see Fig.2) and endows agents with unrealistic behaviors. We build a better simulator (C4) by modifying these behavioral rules, modestly improving human performance to 93.52. Table 5 presents even larger improvements for city-specific simulators (Citi-Gyms), suggesting these are more realistic.

several multi-scale graph convolution modules followed by a fully connected layer to classify the city. We use standard cross-entropy loss to learn this classifier. We get a classification accuracy of $96\%$. This allows us to employ the correct city-specific driver models in our MPC-based planner.

## 5 EXPERIMENTS

**nuPlan Dataset**. We perform all evaluation on the nuPlan MP benchmark (Caesar et al., 2021). The dataset includes driving logs collected in Boston, Pittsburgh, Las Vegas and Singapore, and provides around 10M scenarios, each 20-30 seconds long. nuPlan identifies 73 types of interesting scenarios e.g. {`changing lane`, `starting left turn`, `unprotected right turn`, `etc.`}, but only evalutes 14 in the official benchmark.

**Evaluation Setup and Metrics**. As described in Section 3, the nuPlan benchmark provides three evaluation settings: C1, C2 and C3. In addition, we propose C4, which we claim is more realistic. We evaluate the above mentioned setups using metrics that broadly cover traffic rule violations, human driving similarity, vehicle dynamics and progress towards the goal. The metrics used for *C1* are *ADE* and *FDE*. The metrics used for *C2* and *C3* are *ego progress along expert trajectory*, *speed limit compliance*, *driving direction compliance*, *time to collision within bounds*, and *ego is comfortable*. The overall score tabulated in Table 2- 8 is computed using a weighted average of the above metrics. For more details about the metrics, please refer to (Caesar et al., 2021).

**Implementation Details**. We use the codebases of Dauner et al. (2023) and Caesar et al. (2021) for this work. We modify PDM-C (Dauner et al., 2023) by adding additional longitudinal velocities and lateral offsets to the reference path. In addition, we create additional proposals using hyperparameters such as *min gap to lead agents*, *headway time*, *maximum acceleration* and *maximum deceleration*. In total, our planner generates around 150 proposals per timestep. We train our city classification network for 20 epochs using Adam optimizer Kingma & Ba (2014) with a learning rate of $5e-5$. We pick the radius $R$ for the map data as 100.

**ReactiveReplay-Gym More Closely Resembles the Real-World.** We evaluate the human driver (log) replay planner provided by nuPlan codebase. Human performance on our proposed ReactiveReplay-Gym (which allows agents to follow their ground-truth trajectory and react to other agents) outperforms the Default-Gym (which projects all cars to the road graph) by $0.4\%$ as in Ta-

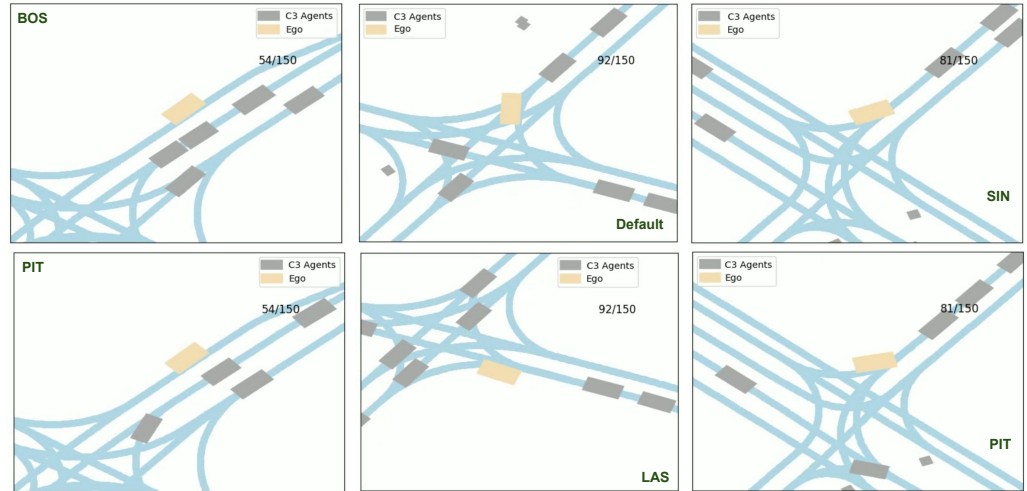

Figure 3: **Agent Behaviours in Different Gyms.** We visualize different agent behaviors (City-Gyms) on three nuPlan scenarios in Pittsburgh. a) Left: Agents tailgate other vehicles more in Boston than in Pittsburgh. This is due to the high acceleration and lower min gap of the BOS-Gym compared to PIT-Gym (cf. Table 4). b) Center: Agents in LAS-Gym have a higher max acceleration compared to agents in the Default-Gym. c) Right: Agents in SIN-Gym have a higher max acceleration and high min gap in compared to PIT-Gym. We include videos of these scenarios in the supplement.

| City | Target Vel. ($\theta_0$) | Min Gap ($\theta_1$) | Headway Time ($\theta_2$) | Max Acceleration ($\theta_3$) | Max Deceleration ($\theta_4$) |
|---|---|---|---|---|---|
| Default ($\theta^{DEF}$) | 10 | 1.0 | 1.5 | 1.0 | 2.0 |
| PIT ($\theta^{PIT}$) | 10 | 2.0 | 0.5 | 1.5 | 3.0 |
| BOS ($\theta^{BOS}$) | 10 | 1.0 | 1.5 | 2.0 | 3.5 |
| SIN ($\theta^{SIN}$) | 10 | 2.5 | 1.5 | 2.0 | 1.0 |
| LAS ($\theta^{LAS}$) | 10 | 1.0 | 0.5 | 1.5 | 0.5 |

Table 4: **Optimizing City-Specific Gyms.** We optimize per-city gyms ($\theta^{CITY}$) using the objective function in Equation 1. The City-Gym models ($\theta^{CITY}$) are optimized to mimic the recorded behaviours of human drivers for that city such that the human ego-vehicle driver achieves high accuracy. These parameters are also used for the City-Driver model.

ble 3. Moreover, we highlight unrealistic agent behaviours of C3 in Figure 2(b) compared to the log replay in Figure 2(a). In contrast, we argue that C4 agents' behaviour in Figure 2(c) matches the log replay.

**Insights from City-Specific Gyms**. We compare the optimized hyperparameters of each city-specific gyms ($\theta^{CITY}$) from Table 4. We present salient observations below:

- Simulated agents in the Default-Gym drive less aggressively than real drivers in PIT and BOS. The Default-Gym has lower *maximum acceleration* and *maximum deceleration* compared to PIT and BOS.

- Drivers in BOS and LAS tailgate other vehicles more aggressively (lower min gap) to PIT and SIN. We can also observe this phenomenon in Figure 3 between BOS and PIT on the left.

- BOS seems to have the most aggressive drivers and LAS seems to have least aggressive ones, as defined by *maximum acceleration* and *maximum deceleration*.

**Ensemble of City-Specific Gyms offers Simulator more Realism**. As discussed in Section 3, a simulator becomes realistic when the Gym mimics recorded behaviour of agents when the ego vehicle is on world-on-rails. From Table 5, we note that ensemble of city-specific gyms (which we call City-Gym) improves over the default-Gym by $0.97\%$.

| Models | City | Gym Models | C4 |
|---|---|---|---|
| Human (Log) Replay Planner | All | ReactiveReplay-Gym | 93.52 |
| | All | City-Gym | **94.49** |

Table 5: **City-Specific Gyms Improve Simulation Realism**. nuPlan evaluates performance in a single world simulator (C3 with Default-Gym), which we show does not accurately capture real-world agent behaviors (Human performance in the Default-Gym and real-world in Figure 1). ReactiveReplay-Gym reactive logic is same as Default-Gym in C3. We create a more realistic world simulator for each of the deployed cities (PIT, BOS, SIN and LAS), and interestingly, find different agent behaviors are appropriate for different cities. Log replay (human) performance is better in city-specific gyms. Additionally, we find that statistics computed from our city-specific world models match those computed from actual city data (Figure 1).

| Models | Driver Models | City | Gym Models | C3 (City-Gym) | C4 (City-Gym) |
|---|---|---|---|---|---|
| PDM-C | Constant Velocity | PIT | PIT-Gym | 90.26 | 89.84 |
| | Constant Velocity | BOS | BOS-Gym | 92.80 | 90.94 |
| | Constant Velocity | SIN | SIN-Gym | 93.58 | 93.55 |
| | Constant Velocity | LAS | LAS-Gym | 94.99 | 93.01 |
| Ours | PIT-Driver | PIT | PIT-Gym | 92.21 | 91.10 |
| | BOS-Driver | BOS | BOS-Gym | 93.52 | 92.05 |
| | SIN-Driver | SIN | SIN-Gym | 94.26 | 94.87 |
| | LAS-Driver | LAS | LAS-Gym | 96.00 | 93.50 |
| PDM-C | Constant Velocity | All | City-Gym | 92.90 | 91.83 |
| Ours | City-Driver | All | City-Gym | **93.99** | **92.88** |

Table 6: **City-Specific Driver Models**. We show the performance of model-predictive control (MPC) algorithms that unroll a (ego) world model. We specifically evaluate the performance of different world models, including a default non-reactive world (where other agents move with constant velocity, as in PDM-C (Dauner et al., 2023)) and various reactive worlds (including Pittsburgh-Driver, Boston-Driver, etc.). We find that City-Driver models consistently improve in performance over prior art across a variety of gym environments, by up to 1.8%.

**PDM-C Baseline Performs Worse on City-Specific Gyms**. In comparison to PDM-C baseline performance on C3 and C4 with Default-Gym (as in Table 3), we notice a drop in performance of $0.71\%$ and $1.19\%$ on C3 and C4 when evaluated on ensemble of city-specific Gyms. As discussed in Section 4, this can be attributed to the constant velocity driver model used by PDM-C which is agnostic to city-level agent behaviours.

**City-Driver Adapts to Different Driving Conditions**. City-Driver is a MPC-based planner that unrolls and executes based on a city-specific world model. It demonstrates an improvement of $1.09\%$ in C3 and $1.05\%$ in C4 over the PDM-C baseline, when evaluated on an ensemble of City-Gyms as tabulated in Table 6. This shows that city-specific reactive driver models yield better planning performance compared to constant-velocity. For example, in order to avoid collisions in Boston, it is important for the ego-vehicle to also drive aggressively.

**City-Gyms Evaluate Planner Robustness**. City-Gyms allow one to evaluate a planner on out-of-distribution behaviors. Interestingly, one can endow the same recorded logs with agent behaviors learned from different cities. For example, we can evaluate PIT-Driver on Pittsburgh driving logs, but endow agents with Boston-like behaviors. We can evaluate the robustness of the motion planners by noting the variance in performance across all gyms. PDM-C and our model achieve a variance of 1.62 and 2.20 respectively. Our model has a higher variance because it is explicitly optimized for PIT. In addition, we note that the PIT-Driver model performs competitively on BOS and SIN while poorly on LAS. This can be due to a large mismatch in *min gap* and *maximum deceleration* between the PIT-Driver and LAS-Gym as shown in Table 4. Future work can address this limitation by optimizing the Driver models at the scenario or agent level.

| Models | Driver Models | City | Gym | C4 (City-Gym) |
|---|---|---|---|---|
| PDM-C (Dauner et al., 2023) | Constant velocity | PIT | PIT-Gym | 89.84 |
| | Constant Velocity | PIT | BOS-Gym | 91.18 |
| | Constant Velocity | PIT | SIN-Gym | 91.73 |
| | Constant Velocity | PIT | LAS-Gym | 88.92 |
| Ours | PIT-Driver | PIT | PIT-Gym | 91.10 |
| | PIT-Driver | PIT | BOS-Gym | 90.77 |
| | PIT-Driver | PIT | SIN-Gym | 91.82 |
| | PIT-Driver | PIT | LAS-Gym | 88.39 |

Table 7: **Robustness Evaluation**. To robustly evaluate planners, we advocate a strategy where given planners are evaluated against different agent behaviors, such as city-specific gyms (PIT-Gym, BOS-Gym, etc.) for each city. The variance of PDM-C and our model are 1.62 and 2.2 respectively, indicating that our model is less robust compared to PDM-C since it is explicitly optimized for PIT. This limitation can be addressed by optimizing the Driver models for each scenario or agent.

| Models | Driver models | City | C2 (Default) | C3 (Default) |
|---|---|---|---|---|
| PDM-C | Constant Velocity | All | 94.49 | 93.61 |
| Ours | City-Driver | All | **94.97** | **95.13** |
| Human | NA | All | 97.6 | 93.12 |

Table 8: **nuPlan benchmark results**. Planners are evaluated on the nuPlan test set on C2 and C3 metrics. Importantly, our City-Driver models dramatically improve over prior art on the default nuPlan benchmark, reducing error from 6.4% to 4.8%. The online benchmark test set was not active at the time of the paper submission. We attach additional visuals comparing our planner to PDM-C in the supplement.

**City-Driver Achieves SoTA Performance**. We evaluate our proposed MPC-based planner with city-specific driver models on C2 and C3 metrics in nuPlan benchmark. To ensure parity with prior work, we use nuPlan's Default-Gym. City-Driver outperforms PDM-C by $0.48\%$ and $1.52\%$ in C2 and C3 respectively. We can infer that city-specific MPC-based planners which adapt to agent behaviours can perform well on planning benchmarks. Future work can naturally extend our city-level gym and driver models to optimize for particular scenarios and individual agents, further improving simulator realism and pushing ahead planning benchmarks.

## 6    CONCLUSION

In this paper, we evaluate the quality of nuPlan's simulator and find that it does not accurately reflect real-world human behavior. We propose alternate city-specific reactive world models to evaluate planning performance and demonstrate that these simulated environments more closely match log-replay. Lastly, we present City-Driver, an MPC-based planner that unrolls and executes a city-specific world model that adapts to different driving conditions and achieves state-of-the-art performance on the nuPlan closed-loop reactive benchmark.

**Limitations.** Since we use the IDM's PID controller in both our gym and driver models, we inherit the PID's flaws. We note that the IDM can be too conservative (when it mistakes the parked vehicle as lead vehicle and stops) or too aggressive (when traveling at high speeds along a curved road. Secondly, the IDM does not interact with other agents except the lead vehicle. Therefore, this may lead to lateral collisions as the PID controller only provides longitudinal velocity control. The IDM parameters are optimized at the city level performance on the humans and not at agent level. Hence, these set of parameters can simulate scenario where the agents can collide (For *eg.* min gap parameter as 0.5m or 1m and we have a long vehicle like truck as lead agent to the ego).

**Future Work.** Although we primarily focus on building city-specific gyms and driver models using the IDM, learning-based approaches can potentially yield better results. The human performance on our proposed City-Gyms is still only $94.49\%$. This shows that simulator realism can be further improved by optimizing our proposed metric per-agent.

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
